# Associations Between 10-Year Physical Performance and Activities of Daily Living Trajectories and Physical Behaviors in Older Adults

**DOI:** 10.3390/ijerph22050704

**Published:** 2025-04-29

**Authors:** Mikael Anne Greenwood-Hickman, Weiwei Zhu, Abisola Idu, Laura B. Harrington, Susan M. McCurry, Andrea Z. LaCroix, Pamela A. Shaw, Dori E. Rosenberg

**Affiliations:** 1Kaiser Permanente Washington Health Research Institute, Seattle, WA 98101, USA; weiwei.zhu@kp.org (W.Z.); abisola.idu@kp.org (A.I.); laura.b.harrington@kp.org (L.B.H.); pamela.a.shaw@kp.org (P.A.S.); dori.e.rosenberg@kp.org (D.E.R.); 2Division of Health Systems Science, Kaiser Permanente Bernard J Tyson School of Medicine, Pasadena, CA 91101, USA; 3School of Nursing, University of Washington, Seattle, WA 98195, USA; smccurry@uw.edu; 4Herbert Wertheim School of Public Health and Human Longevity Science, University of California, San Diego, La Jolla, CA 92093, USA; alacroix@health.ucsd.edu

**Keywords:** physical activity, sedentary behavior, sitting, sleep, physical function

## Abstract

Physical function is likely bidirectionally associated with physical activity (PA), sedentary behavior (SB), and sleep. We examined trajectories of physical function as predictors of these behaviors in community-dwelling adults aged ≥65 y without dementia from the Adult Changes in Thought cohort. Exposures were trajectories of physical performance (short Performance-Based Physical Function [sPPF]) and self-reported activities of daily living (ADL) impairment. Outcomes were device-measured PA and SB and self-reported sleep. We fit linear mixed-effects models to define trajectory slopes and intercepts for each functional measure over the prior 10 years. We used multivariable linear regression to investigate the relationship between trajectory features and outcomes, using bootstrap confidence intervals. Participants (N = 905) were 77.6 (SD = 6.9) years old, 55% female, 91% white, and had a median sPPF score of 9 (IQR = [8, 11]) and median impairment of 1 ADL (IQR = [0, 2]) at the time of activity measurement (baseline). Steeper decreases in sPPF (0.3-unit, 25% of the range) were associated with fewer steps (−1180, 95% CI = [−2853, −185]) and less moderate-to-vigorous PA (−15.7 min/day [−35.6, −2.3]). Steeper increases in ADL impairment were associated with 35.0 min/day (4.3, 65.0) additional sitting time, longer mean sitting bout duration (3.5 min/bout [0.8, 6.2]), fewer steps (−1372 [−2223, −638]), less moderate-to-vigorous PA (−13 min/day [−22.6, −5.0]), and more time-in-bed (25.5 min/day [6.5, 43.5]). No associations were observed with light PA or sleep quality. Worsening physical function is associated with lower PA and higher SB, but not with light-intensity movement or sleep quality, supporting the bidirectional nature of the relationship between physical function and physical behaviors.

## 1. Introduction

The maintenance of physical function as we age is critical to the preservation of the ability to engage in both physical and social activities, remain independent in activities of daily living (ADLs), and to preserve overall quality of life in later years [1]. The physical behaviors we engage in throughout the day—sleep, sedentary behavior (SB), and physical activity (PA)—likely contribute to the preservation of physical function across the life course, both independently and synergistically [2]. However, most older adults spend the majority of their waking hours engaged in SB and few meet physical activity guideline recommendations [3,4,5]. These trends are often accentuated for older adults with chronic conditions and functional limitations [4,6,7]. Sleep quality and duration also tend to decline with age [8,9]. Despite these concerning trends in older adult physical behaviors and the likelihood of bidirectionality, to date, most research has focused on the impacts of each of these behaviors on physical function outcomes and has not explored how declines in function may be driving changes in physical behaviors.

The current literature examining the relationships between PA, SB, and functional health in older adults suggests a strong association between higher levels of moderate-to-vigorous intensity PA (MVPA) and physical function [10]. Studies have also suggested that higher daily step counts, a measure of total daily PA, are associated with better self-reported and objective measures of physical functioning [11,12,13]. Any independent role of SB and sitting patterns in affecting physical function, however, is not yet well characterized. Evidence is growing that links total sedentary time and patterns of sedentary time to physical function, but much of this evidence is cross-sectional, leaving ambiguity about whether high SB and low PA drive decreases physical function, or whether physical function impairment drives low activity and high SB [14,15,16,17]. Data in the Adult Changes in Thought (ACT) cohort, a US-based longitudinal cohort of older adults described in more detail in Section 2, demonstrated that higher mean sitting bout duration and sitting time were associated with slower gait speed, chair stand time, and lower combined physical function scores cross-sectionally [18]. There is potential for reverse or bidirectional causality in associations between PA, SB, and physical function, but, to our knowledge, no studies to-date have evaluated historic trajectories in physical function as a predictor of future PA and SB, either individually or in combination.

The existing literature on the relationship between sleep and physical function is predominantly cross-sectional, and findings are mixed. Three cross-sectional studies [19,20,21] and one prospective analysis [22] have suggested that higher sleep quality is associated with better objective measures of physical performance, fewer impairment to activities of daily living (ADLs), and less pain and disability in older adults. Sleep duration has also been explored cross-sectionally in relation to physical function in later life, with some studies finding no association [19], and others finding relationships with both long and short sleep duration [23,24], frailty, and physical function. Notably, these studies consider sleep quality and duration as predictors of physical function. We are not aware of any studies that explore the reverse relationship, namely between historical physical function and how it predicts sleep quality or duration in later life. It is plausible that worse physical function, with associated pain and disability, could lead to long-term sleep disturbance [25,26].

Fully characterizing the relationship between physical function and physical behaviors is critical to optimizing the timing and approach to future health and activity promotion interventions for older adults. To our knowledge, no studies have examined longitudinal trajectories of physical function in relation to PA, SB, and sleep, a critical gap in our understanding of the complex relationship between physical function and these behaviors. We aim to define trajectories of objective physical performance and self-reported impairment to ADLs and examine if these trajectories predict device-measured activity (PA and SB) and self-reported sleep among members of the ACT cohort. We hypothesized that more stable trajectories of both physical performance and functional capacity (i.e., better current scores and less decline prior to baseline) would be associated with higher PA, lower SB, and better self-reported sleep quality.

## 2. Materials and Methods

### 2.1. Adult Changes in Thought (ACT) Cohort

We used data from the Activity Monitoring sub-study of the Adult Changes in Thought (ACT) study, which has been described previously [4]. Briefly, ACT is an ongoing longitudinal cohort study of older adults that began in 1994 and maintains an active enrollment of approximately 2000. ACT participants are recruited from random samples of the membership panels of Kaiser Permanente Washington aged 65 and above who are free from dementia at enrollment. Participants are followed every 2 years until death, disenrollment, or onset of dementia. Starting in 2016, ACT began collecting device-based and self-reported assessments of PA, SB, and sleep from eligible and consenting members of the cohort to measure physical behavior. Participants were invited to participate in device data collection if they were not wheelchair bound, receiving hospice or care for a critical illness, residing in a nursing home, and if no memory problems became evident during testing. A total of 1885 ACT participants met these criteria and were approached to participate in the first wave of device data collection. Those choosing to participate provided full written informed consent to all study procedures. Only data from the initial wave of device data collection, from 2016 to 2018, were included in these analyses. All data collection procedures were reviewed and approved by the Kaiser Permanente Washington (KPWA) institutional review board.

### 2.2. Physical Performance Exposure

Objective physical function was assessed by three in-person standardized physical performance tasks during the ACT biennial study visits: gait speed as measured by the average of two 10-foot timed walks; chair stand time (time needed to move from a seated position in a chair to a standing position, repeated five times); and grip strength as measured by handheld dynamometer (average of three attempts in the dominant hand). As in prior ACT studies, each task was scored between 0 and 4 based on sex-specific cutpoints. Scores on each task were summed to construct a 12-point short Performance-Based Physical Function (sPPF) score (range 0–12), with higher scores indicating better physical function. A score of 0 for a given task indicated inability to complete that task [27,28]. ACT’s original Performance-Based Physical Function (PPF) test (Cronbach α = 0.74) included an additional balance task and correlated with self-reported level of difficulty performing ADLs [27]. However, the balance task was dropped from the study’s standard assessments and was unavailable for these analyses.

### 2.3. Activities of Daily Living (ADL) Exposure

Self-report of subjective difficulty with 16 distinct ADLs was collected at each biennial ACT study visit (e.g., bathing/showering, light housework, walking up a flight of stairs, getting out of bed, and walking one-half mile). Participants endorsed the level of difficulty (none, some, a lot, unable to do) experienced with each activity. All ADL tasks endorsed as having some difficulty or more summed to produce a summary score ranging from 0 to 16. If any of the 16 items were missing, the sum was set to missing. ACT’s original 17-item ADL summary measure was previously described [29] and comprises items from several validated measures of self-reported physical function in older adults [30,31,32,33]. As of 2008, the item about ability to reach above one’s head was dropped from the original 17-item scale, leaving the 16 items summarized for this analysis.

### 2.4. Physical Behavior Outcome Variables

Details of ACT device data collection protocols were outlined previously [20]. Briefly, we measured sitting time, standing time, stepping time, and step counts with the activPAL accelerometer. Participants wore the activPAL 3 micro (PAL Technologies, Glasgow, Scotland, UK) at the front center of their upper thigh for 7 days using a 24 h wear protocol. We used proprietary PAL Technologies software (v7.2.38.2, VANE processing algorithm) to extract event-level files. Within each event file, consecutive activities of the same activity type (i.e., sitting, standing, stepping) were combined, and in-bed time based on participant self-reported logs were removed to compute metrics only for waking (out-of-bed) hours. ActivPAL summary measures included the following: mean daily total sitting time (min/day), mean daily total standing time (min/day), mean daily total sit-to-stand transitions (number of transitions/day), mean sedentary bout duration (minutes/sedentary bout), and mean daily total steps.

Light-intensity and MVPA were measured with the ActiGraph wGT3X+ accelerometer (ActiGraph LLC, Pensacola, FL, USA) worn at the hip for 7 days using a 24 h wear protocol. Raw ActiGraph data were collected at 30 Hz and were processed into a proprietary count variable at 15 s epochs using the “normal” filter in ActiLife software (v 6.13.3). Cutpoints calibrated for older adults developed in a Women’s Health Initiative laboratory study were applied to the data. Specifically, intensity classifications using vector magnitude counts per 15 s epoch were as follows: ≤18 for sedentary time, 19–518 for light-intensity PA (LPA), and >518 for MVPA [34]. Daily summary ActiGraph measures included the following: mean daily total time engaged in LPA (min/day) and mean daily total time engaged in MVPA (min/day). For both activPAL and ActiGraph measures, a minimum of 4 days (regardless of weekday vs. weekend day) with 10 or more hours of waking wear time, as defined by the presence of valid device data during participant self-reported awake periods, was required on both devices to be included in analyses with outcomes from that device.

Sleep time was approximated using time-in-bed from self-reported diaries kept by participants during accelerometer wear. Participants recorded the time they got into bed and out of bed for each day they wore the devices. We also captured self-reported sleep quality with the 8-item PROMIS sleep disturbance scale (Form 8a) [35,36]. Scores are converted to T-scores with a mean of 50 and a standard deviation of 10. Higher scores indicate higher levels of sleep disturbance and lower quality sleep.

### 2.5. Demographic and Health Characteristics

Participants self-reported the following demographic and health characteristics at their ACT study enrollment visit: gender (male/female), race (American Indian or Alaskan Native, Asian, Black, Native Hawaiian or Pacific Islander, White, or Other), Hispanic ethnicity (yes/no), and years of education (dichotomized as ≥16 vs. <16 years). Additionally, time-varying participant characteristics were collected by participant self-report at the most recent ACT visit, generally within 1 year prior to physical behavior measurement: status of currently working for pay (yes/no), living arrangement (living alone/with others), depressive symptoms using the Center for Epidemiologic Studies Depression Scale (CES-D; continuous, dichotomized as <10 vs. ≥10) [37], and self-rated health (4 levels, fair/poor vs. good/very good/excellent). Additional characteristics were objectively assessed at the ACT visit within 1 year prior to physical behavior measurement: body mass index (BMI), cognitive function (CASI) bifactor score (see Appendix A for more detail), total activPAL daily wear time (for activPAL outcomes only), and total Actigraph daily wear time (for Actigraph outcomes only). The Charlson Comorbidity Index was computed using ICD-10 codes from the participants electronic medical record over the year prior to device wear.

### 2.6. Statistical Methods

We provide descriptive statistics for the baseline covariates and descriptive characteristics. The nine outcomes of interest were as follows: time sitting, time standing, time stepping, steps per day, and average bouts of sitting from activPAL; light PA and MVPA from ActiGraph; self-reported time-in-bed; and PROMIS sleep disturbance score. All activity time measures were analyzed as min/day.

For each of the nine activity behavior outcomes, we examined the adjusted association between these outcomes and subject-specific trajectory characteristics (individual-level slope and intercept from a fitted linear mixed effects model) for our primary exposures, ADL, and Performance-Based Physical Function (sPPF) using multivariable linear regression. For comparability across the physical function assessment score models, only individuals with at least one observation for both the ADL and sPPF instruments in the 10 years prior to the activity device-wear visit and complete covariates were eligible for the analytical cohort (Appendix A). We first fit linear mixed effects models (LMMs) to separately analyze longitudinal data for the ADL and sPPF scores collected over 10 years prior to the 2016–2018 device wear. We evaluated a model with a random intercept and one with both a random intercept and slope. The model selection process was guided by the Akaike information criterion (AIC), where a lower AIC value determines the better model. Ultimately, we selected the random intercept and slope model and derived the best linear unbiased prediction (BLUP) estimates for the subject-specific slopes and intercepts to describe the trajectories for the sPPF and ADL scores. We set time zero as the start of the activity monitor measurement time, allowing us to interpret the model intercept as the summary score at baseline. Next, we fit linear regression models to investigate the relationship between these trajectory features and the outcomes of interest, treating the subject-specific intercept and slope estimates as exposure variables. Due to the concern that the intercept is an intermediate variable on the causal pathway for the effect of the trajectory slope on the activity behavior outcome, we considered two multivariable linear regression models: one including only the trajectory intercept and one including the trajectory slope. In all linear mixed effects trajectory and outcome linear regression models, we adjusted for age (continuous), gender, education years, current work status, living arrangement, self-rated health, body mass index (continuous), CESD score (continuous), and CASI-IRT (continuous) at the time of physical behavior measurement. To address the uncertainty associated with the estimated trajectories derived from the first step, we used the bootstrap percentile method to calculate the 95% confidence interval (CI) for the coefficients estimated from the multivariable linear regression model, with 1000 bootstrap replications. Naïve standard errors are also presented for illustrative purposes.

Statistical analyses were conducted in SAS 9.4 and R (4.3.2), utilizing the ‘lme4’ package for Linear Mixed Effect Models. In regression analyses, we employed a complete case approach and observations with missing data were excluded. As a sensitivity analysis to explore the impact of selection bias on primary analyses, we built a logistic regression selection model and performed an inverse-probability weighted (IPW) regression analyses to account for the potential selection bias introduced by missingness, using the same bootstrapped confidence interval approach described above. Unless otherwise stated, an alpha level of 0.05 is used for statistical significance; for bootstrapping methods, we examined whether the 95% confidence interval excluded 0.

## 3. Results

Of the 1885 invited to participate in device data collection, 951 individuals wore both devices for at least 4 valid wear days, and 905 of the 951 (95.2%) had at least one measure of both the sPPF and ADL measures in the 10 years prior to the device-wear baseline and complete data for the regression covariates (Appendix A). Appendix A compares the demographics of those in the analytic cohort vs. those invited to wear a device who did not participate. Compared to nonparticipants, those included (Table 1) were on average younger, with a mean age of 77.6 (SD 6.9) vs. 81.3 (SD 8.1) years, more educated (75% with ≥16 years of education vs. 66%), less likely to live alone (34% vs. 43%), and slightly less diverse on ethnicity and race. Table 1 describes demographic and clinical characteristics of the analytic cohort at the time of physical behavior measurement. Most participants in the analytic cohort identified as white race (91%), had at least 16 years of education (75%), and self-reported being in good health or better (93%), with a median (IQR) Charlson Index of 0 (0, 2) and average (SD) BMI of 26.9 (4.8) kg/m^2^. The median (IQR) for ADL score was 1 (0, 2) and for sPPF was 9 (8, 11). Participants with lower sPPF (<9) at baseline physical behavior measurement were older (mean age 80.3 years vs. 76.0 years), had a higher percentage of women (40% vs. 47%), a lower percentage of more than 16 years of education (67% vs. 79%), and higher percentages of fair/poor self-rated health (13% vs. 4%), depressive symptoms (11% vs. 7% with a CES-D score >10), and higher Charlson Comorbidity scores (mean score 1 vs. 0). Similar patterns were noted for participants with ADL impairment (1+ vs. 0) at baseline (Appendix A).

Appendix A shows the actual trajectories of ADL and sPPF in the 10 years prior to device wear for 20 randomly selected participants with at least two measures for both physical function scores. In this figure, the two physical function scores tend to track together, with an increase in ADL impairment being associated with a decline in sPPF. At baseline, ADL and sPPF score had a Pearson’s correlation coefficient of −0.47.

### 3.1. Physical Performance Trajectories

Figure 1 shows the fitted sPPF trajectories by quartile of the slope. There was a moderate level correlation (0.42) between the random intercept and slope random effects. Unadjusted associations between sPPF trajectory slope and physical behavior outcomes of interest are displayed in Figure 2 (plots of unadjusted trajectory intercept are available in Appendix A). In general, both intercept and slope have similar trends for each outcome of interest in terms of direction and magnitude, with lower sPPF intercept and slope values trending towards more sitting, longer mean sitting bout duration, and lower steps, stepping time, and MVPA.

Trajectories are estimated from an adjusted mixed model and shown for the baseline group: female, aged 78, with a minimum of 16 years of education, retired, no signs of depression, resides with others, in good health, body mass index (BMI) of 27, and standardized Cognitive Abilities Screening Instrument (CASI) score of 0.

Table 2 includes adjusted results for the multivariable linear regression models including sPPF trajectory intercept and slope exposures for a 1-unit decrease in the intercept (baseline value) and 0.3-unit decrease in the slope of the sPPF score trajectory, where a slope of 0.3 represents a change of 3 points (25% of the range) over 10 years (a decrease represents a decline in physical functioning). A unit change in the sPPF trajectory intercept was associated with more sitting time (+6.6 min/day 95% Confidence Interval: [1.2, 11.7]), shorter stepping time (−4.1 min [−5.7, −2.6]), fewer daily steps (−359 steps/day [−502, −226]), longer mean sitting bout duration (+0.4 min/bout [0.05, 0.8]), and less daily MVPA (−4.4 min/day [−6.1, −3.0]). A 0.3-unit change in the sPPF trajectory slope was associated with fewer daily steps (−1180 steps/day [−2853, −185]) and less daily MVPA (−15.7 min/day [−35.6, −2.3]). We did not observe associations in either sPPF trajectory characteristic (slope or intercept) with daily standing time, daily LPA, time-in-bed, or sleep quality. Appendix A presents the same model estimates, but with confidence intervals derived from the naïve regression model-based standard errors that do not account for the uncertainty in the BLUP exposures; conclusions were qualitatively the same in all cases except the min/day stepping, where the confidence interval excluded 0 only for the naïve version of the confidence interval. Again, the confidence intervals based on the naïve standard errors (Appendix A) were naarrower. Appendix A shows the results of the IPW analysis, which accounted for the missing data. The AUC for the fitted selection model equaled 0.71, with estimated weights ranging from 1.2 to 33, indicating some success in capturing the reasons for missingness. Results from the IPW regressions were very similar with respect to the estimated effect size and, as expected, had slightly wider confidence intervals due to the weighting. Due to the larger variance two associations (sitting bouts with the baseline and MVPA with the slope) went from borderline significant to borderline non-significant.

### 3.2. Activities of Daily Living Trajectories

Figure 3 shows the fitted ADL trajectories by quartile of the slope, which shows the relatively strong correlation between the intercept (baseline) ADL and slope (0.77). Unadjusted associations between ADL trajectory slope and physical behavior outcomes of interest are displayed in Figure 4 (plots of unadjusted trajectory intercept are available in Appendix A). In general, both intercept and slope have similar trends for each outcome of interest in terms of direction and magnitude, with higher ADL impairment intercept and slope values trending towards more sitting, longer mean sitting bout duration, more time-in-bed, and lower step count, stepping time, and MVPA. Table 3 shows the adjusted association between a 1-unit increase in the intercept of the subject-specific ADL trajectory (i.e., a 1-unit increase in their baseline ADL summary score) with PA, SB, and sleep outcomes.

Trajectories are estimated from an adjusted mixed model and shown for the baseline group: female, aged 78, with a minimum of 16 years of education, retired, no signs of depression, resides with others, in good health, body mass index (BMI) of 27, and standardized Cognitive Abilities Screening Instrument (CASI) score of 0.

Table 3 also shows adjusted associations between a 0.4-unit increase in the ADL slope (representing a difference across 25% the range) over 10 years (an increase represents more impaired activities of daily living) with PA, SB, and sleep outcomes. A 1-unit increase in baseline ADL impairment was associated with more sitting time (+8.8 min/day [2.9, 13.9]), less standing time (−5.1 min/day [−9.2, −0.01]), less stepping time (−3.7 min/day [−5.2, −2.2]), fewer daily steps (−348 steps/day [−478, −225]), longer mean sitting bout duration (+0.8 min/bout [0.3, 1.3]), less daily MVPA (−3.3 min/day [−5.0, −1.7]), and more daily time-in-bed (+4.5 min/day [1.0, 8.2]). In the adjusted slope models, associations with several of the same outcomes were found. A 0.4-unit increase in the slope of the ADL score was associated with more sitting time (+35.0 min/day [4.3, 65.0]), less stepping time (−14.4 min/day (−24.2, −5.6]), fewer steps per day (−1372 [−2223, −638]), longer sitting bout duration (+3.5 min/bout [0.8, 6.2]), less daily MVPA (−13.0 min/day [−22.6, −5.0]), and more daily time-in-bed (+25.5 min/day [6.5, 43.5]). We did not observe associations in either ADL trajectory characteristic (slope or intercept) with daily standing time, daily LPA, or sleep quality. Appendix A presents the same model estimates, but with the confidence intervals based on the naïve regression model-based standard errors that do not account for the uncertainty in the BLUP exposures; conclusions were qualitatively the same in all cases, though the confidence intervals based on the naïve confidence intervals were narrower, as expected. Appendix A shows the results of the IPW regression analysis, which accounted for the missing data. As noted previously, the AUC for the fitted selection model equaled 0.71, indicating some success in capturing the reasons for missingness. IPW results were again very similar; however, in the IPW analysis, increases in ADL baseline value and slope were also significantly associated with shorter standing time.

## 4. Discussion

We illustrated that declining physical function over 10 years, as measured by both performance-based assessment and self-reported ADL impairment, was associated with lower PA and longer sitting time. Both the intercept (i.e., the cross-sectional relationship at the time of device wear) and the slope (i.e., the rate of decline over the prior 10 years) appeared to be associated with SB, MVPA, and steps, though effect sizes were modest, with relatively large changes in physical function over 10 years needed to impart clinically meaningful changes in activity level. Our results do not indicate strong or consistent associations between physical function trajectory and self-reported sleep time or quality, nor were there observed associations with light intensity PA. Taken together, these findings suggest that declines in physical function result in slightly less movement and longer sitting time. One possibility is that individuals may substitute more extended sitting for higher-intensity movement as physical function declines, but future research is needed to explore this hypothesis.

There were more marked differences (per unit change) in associations between both slope and intercept measures for the historical ADL summary measure than were observed for the objective sPPF measure. It is possible that declines in physical function that rise to the level of limiting ADLs, whether due to objective or perceived impairment, may be more impactful on both sitting and PA behavior than the smaller or more subtle declines in physical performance captured in an objective physical performance measure like the sPPF. Importantly, one component of the sPPF, grip strength task, is unlikely to be strongly associated with the gross motor movements of the physical behaviors explored here. When you further consider that grip strength task replaced a balance task more common to this type of objective physical performance measurement, this may have resulted in an underestimation of observed associations, particularly for lower-intensity activity outcomes. Additionally, the self-reported ADL summary measure is likely to reflect factors such as cognition, mental health, and other non-physical factors that could impact ADL performance beyond an individual’s objective physical performance ability [38,39]. Declines in sPPF were only significantly associated with fewer steps and less MVPA, indicating that these subtler changes in walking speed and lower extremity strength are most impactful on higher intensity PA behavior rather than light PA, sedentary behavior, or sleep.

To our knowledge, this is the first study of physical function trajectories as a predictor of PA, SB, and sleep in older adults. Prior studies, including prior analyses in the ACT cohort, have suggested that higher PA, particularly MVPA and steps, may be a predictor of better physical function in older age groups [11,12,13,18], but the reverse has not been well explored. Similarly, total SB and SB pattern measures have been previously explored as predictors of physical function, suggesting that more sitting and patterns of longer, less interrupted sitting bouts throughout the day are associated with lower levels of both objective and self-reported physical function [14,15,16,17,18]. In our analysis, we observed associations between both sPPF and ADL impairment and SB time and long sitting patterns that were of borderline significance and modest effect size. With respect to activity behaviors, however, we observed that a 3-point change in sPPF or a 4-point change in ADL difficulty over the prior 10 years was associated with over 1000 fewer steps and around 15 fewer minutes of MVPA at the time of activity measurement, but not with lower intensity movement (LPA, standing). While these are clinically remarkable decreases in higher intensity physical activity behaviors, it should be acknowledged that these were associated with relatively large declines in function. It may be helpful to consider the changes in activity behavior associated with a change of half that magnitude (i.e., a 1.5-point decrease in sPPF or a 2-point increase in ADL) over a 10-year period by simply halving the observed estimates. This translates to a decrease of approximately 500 fewer steps and 7–8 min of MVPA per day, which translates to clinically impactful changes in behavior, particularly if considered over the course of a week or month. Because higher intensity PA has the strongest associations with future health outcomes, including mortality and health span [40,41,42,43], even modest decreases may be important to future health.

Furthermore, these analyses found no associations between physical function trajectories and sleep quality and inconsistent findings related to time-in-bed, suggesting either null associations or modest associations requiring larger sample sizes to detect. With exposure and outcome reversed, prior work has suggested that poor sleep quality is associated with poorer physical function [19,20,21,22,23,24]. However, little evidence was identified in the literature exploring physical function status or trajectory as a predictor of future sleep quality. Declines in physical function are typically driven by declines in health status, which led us to carefully control for self-reported health, cognition, and depressive symptoms in these analyses and may explain the null results. Our sample also had relatively average reported time-in-bed with a low prevalence of subjectively reported sleep disturbance, which does not represent the full range of variability in sleep quality among older adults and may have limited our ability to detect an association. Given the limitations in the sleep measures available here, we should interpret these null findings with caution, as small associations may exist that we could not detect here. Future work that leverages objective measures of both sleep duration and quality, and considers alternative modeling frameworks, is warranted to more clearly characterize the relationship between sleep and physical function in older adults.

The preservation of physical function is critical to the preservation of independence and quality of life as we age [1,44,45]. However, to design and optimally time the delivery of health promotion and preservation strategies for older adults to protect physical function, it is imperative to fully characterize its bidirectional link with physical behaviors. In other words, we must understand not just how physical behaviors drive physical function, but also how physical function declines drive activity. In the context of prior literature, the findings from this analysis support the bidirectional nature of the relationship of physical function and physical behaviors, underscoring that the two likely work synergistically to either promote or deteriorate health and independence in older age. In a practical sense, these findings highlight the need for intervention to improve and maintain physical activity and reduced sedentary behaviors early in the life course as well as at times of transition in functional capacity (e.g., after an injury or change in health status). As prior literature has underscored, higher PA and lower SB are protective of future physical function [11,12,13,18]. While age- and morbidity-related declines in physical function will not all be preventable, early intervention in physical activity may maximize functional resilience over time and prevent spiraling declines in future activity and behaviors.

### Strengths and Limitations

This study has several strengths. We investigated objective assessments of PA and SB time and patterns, which are known to be more accurate than self-report [46,47,48]. We were able to explore both an objective measure of physical function and a measure of self-reported functioning in daily life, which are highly related but distinct constructs of older adult physical function, in relation to our physical behavior outcomes. Additionally, more than 15% of included participants were over age 85 years at the time of device wear. Finally, we used methods that account for the uncertainty in the estimated BLUP physical function trajectory exposure measures. This is an often-overlooked step in modeling that uses predicted exposures that can lead to inaccurate confidence interval estimates when neglected [49].

These analyses also have several important limitations. Compared to the entire ACT cohort, our analytic study population had more education, less diversity in race/ethnicity, and appeared healthier on several measures. This limits our interpretation of these findings to more diverse populations. Because participants in device wear were healthier, we may not capture the full range of physical function impairment in the ACT cohort. We did perform a sensitivity analysis to weigh our observed data to better represent original members of ACT cohort invited to participate, and the findings were similar. The predictive accuracy of the selection models (AUC = 0.71) provided some assurance that the IPW sensitivity analyses directly addressed the identified issue of selection bias into the accelerometer sub-study, improving the representativeness of our results; however, we cannot rule out the possibility of residual confounding. Given the limitations of our sample, we were unable to investigate how these associations may differ in key subgroups (e.g., men vs. women, younger vs. older, etc.). Such investigations are warranted in larger samples that provide adequate power. Our objective measure of physical function, the sPPF, includes a grip strength task rather than a balance task typical of the Short Physical Performance Battery (SPPB) [50]. Grip strength is likely to be less sensitive than a balance task to the lower extremity functional changes most relevant to walking and daily activity, making the sPPF less sensitive to such changes overall. These findings with the sPPF may represent conservative estimates of the true associations with physical behaviors and should be compared to studies using the more common SPPB with caution given this difference. Future studies should explore associations with the SPPB. Importantly, while we had objective measures of waking activity, these analyses had no objective measures of sleep time or quality, and the included self-report measures may have inadequate sensitivity, particularly for a sample with high quality sleep and minimal sleep disturbance. Future studies should replicate findings with objectively assessed sleep duration and quality measures. Additionally, these analyses explore associations between physical function trajectories and physical behaviors independently, but there is growing recognition in the field that these behaviors are interrelated. Future studies that explore physical behaviors in combination across the 24 h day are warranted.

## 5. Conclusions

Particularly for measures of self-reported ADL impairment and functioning, both the current level of impairment and the rate of preceding decline appear to be important predictors of PA and, to a lesser extent, sitting time and patterns in older age. When contextualized among the existing literature supporting PA and SB as predictors of physical function, our findings support the plausibility of bidirectional relationships between physical function and movement behaviors, especially higher-intensity movement, in older age. Light-intensity movement and sleep behaviors, however, may be less impacted by current and prior physical function trajectories, which is reassuring. Given this observed bidirectionality, however, these findings underscore the need for early intervention to promote physical activity and reduce sedentary behaviors earlier in the life course before functional declines appreciably begin. Such early intervention may have a compounding effect, preventing functional declines that may then beget future declines in both activity and function.

## Figures and Tables

**Figure 1 ijerph-22-00704-f001:**
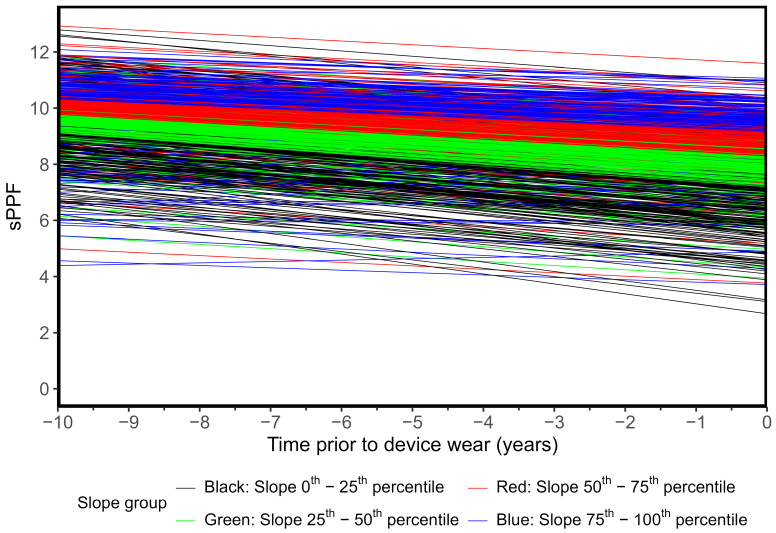
Estimated trajectories for the short Performance-Based Physical Function (sPPF) score by quantiles of individual-specific fitted slope (N = 905).

**Figure 2 ijerph-22-00704-f002:**
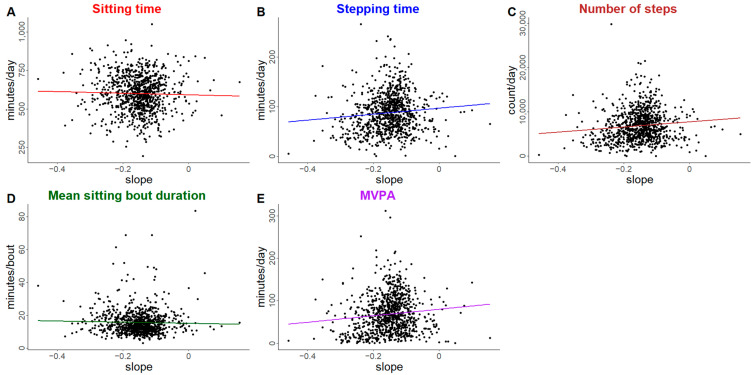
Unadjusted association between sPPF trajectories with physical behavior outcomes of interest. Panels represent each of the following outcomes of interest: (**A**). Average daily sitting time (minutes/day), (**B**). Average daily stepping time (minutes/day), (**C**). Average number of steps per day (steps/day), (**D**). Mean sitting bout duration (minutes/bout), (**E**). Average daily moderate-to-vigorous physical activity (minutes/day). Only outcomes with a statistically significant association with the physical function trajectory in adjusted models are displayed.

**Figure 3 ijerph-22-00704-f003:**
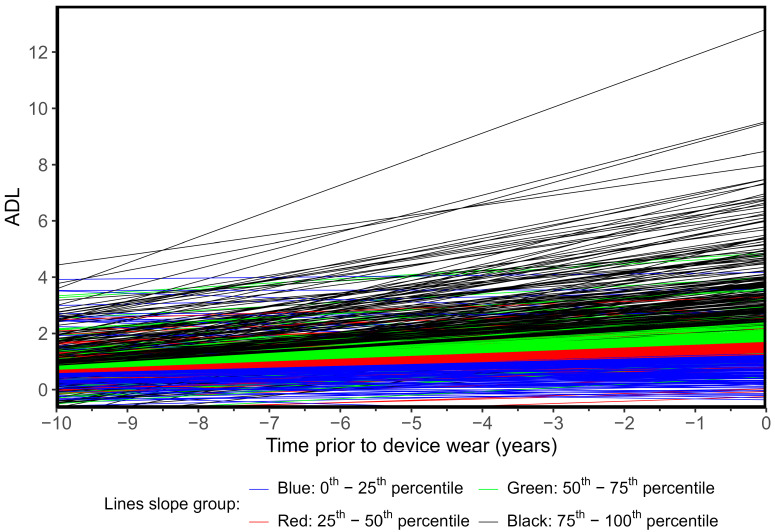
Estimated trajectories for Activities of Daily Living (ADL) summary score by quantiles of individual-specific fitted slope (N = 905).

**Figure 4 ijerph-22-00704-f004:**
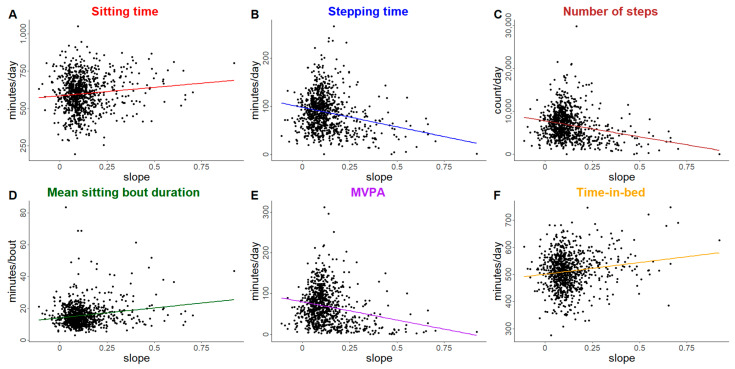
Unadjusted association between ADL trajectories with physical behavior outcomes of interest. Panels represent each of the following outcomes of interest: (**A**). Average daily sitting time (minutes/day), (**B**). Average daily stepping time (minutes/day), (**C**). Average number of steps per day (steps/day), (**D**). Mean sitting bout duration (minutes/bout), (**E**). Average daily moderate-to-vigorous physical activity (minutes/day), (**F**). Average daily time-in-bed (minutes/day). Only outcomes with a statistically significant association with the physical function trajectory in adjusted models are displayed.

**Table 1 ijerph-22-00704-t001:** Demographic and clinical characteristics at device wear for the Adult Changes in Thought (ACT) study analytic cohort (N = 905).

	Activity Monitor ACT Cohort	sPPF: 0–8	sPPF: 9+
	N = 905 ^a^	N = 334 ^a^	N = 571 ^a^
	N (%) Mean (SD) Median [Q1, Q3]	N (%) Mean (SD) Median [Q1, Q3]	N (%) Mean (SD) Median [Q1, Q3]
**Age (years) category,** n (%)			
65–74	378 (41.8%)	91 (27.2%)	287 (50.3%)
74–84	381 (42.1%)	151 (45.2%)	230 (40.3%)
85+	146 (16.1%)	92 (27.5%)	54 (9.5%)
**Age (years),** mean (SD)	77.6 (6.9)	80.3 (7.4)	76.0 (6.1)
**Gender,** n (%)			
Female	502 (55.5%)	200 (59.9%)	302 (52.9%)
Male	403 (44.5%)	134 (40.1%)	269 (47.1%)
**Race,** n (%)			
Asian	28 (3.1%)	13 (3.9%)	15 (2.6%)
Black	13 (1.4%)	8 (2.4%)	5 (0.9%)
White	820 (90.6%)	291 (87.1%)	529 (92.6%)
Other or mixed **^b^**	43 (4.8%)	22 (6.6%)	22 (3.9%)
**Latino/Hispanic ethnicity**, n (%)	12 (1.3%)	4 (1.2%)	8 (1.4%)
**Currently work for pay,** n (%)	168 (18.6%)	46 (13.8%)	122 (21.4%)
**Education level 16+ years,** n (%)	677 (74.8%)	224 (67.1%)	453 (79.3%)
**Live alone**, n (%)	308 (34.0%)	131 (39.2%)	177 (31.0%)
**Self-rated health,** n (%)			
Excellent	178 (19.7%)	37 (11.1%)	141 (24.7%)
Very good	393 (43.4%)	127 (38.0%)	266 (46.6%)
Good	271 (29.9%)	128 (38.3%)	143 (25.0%)
Fair/poor	63 (7.0%)	42 (12.6%)	21 (3.7%)
**Depressive symptoms CES-D Score ≥ 10,** n (%) **^b^**	77 (8.5%)	37 (11.1%)	40 (7.0%)
**Depressive symptoms (CES-D) ^b^**	2 [1, 5]	3 [1, 6]	2 [0, 5]
**Charlson Comorbidity Index**	0 [0, 2]	1 [0, 2]	0 [0, 1]
**Activity of Daily Living (ADL) score**	1 [0, 2]	2 [1, 4]	0 [0, 1]
**Short Performance-Based Physical Function (sPPF) score**	9 [7, 11]	7 [5, 8]	10 [9, 11]
**CASI score,** mean (SD)	0.6 (1.0)	0.3 (1.0)	0.7 (0.9)
**BMI**, Mean (SD)	26.9 (4.8)	27.5 (5.5)	26.6 (4.3)

Note. Abbreviations: BMI: body mass index, CES-D: total score on the shortened 10-item Center for Epidemiology Studies Depression Scale, CASI: standardized CASI score which was computed from summing each of the nine CASI domain scores, Q: quartile, SD: standard deviation, sPPF: short Performance-Based Physical Function Score (sPPF), ADL: Activities of Daily Living. ^a^ Percentage is calculated out of the column total N = 905 (All), 334 (SPPF: 0–8), 571 (SPPF: 9+). Missing values of sPPF at baseline (n = 32) were imputed, for purposes of this summary table, using the most recent non-missing value within the 10-year window prior to baseline. The following variables had missing data: race n = 1, Latino/Hispanic ethnicity n = 3, Charlson Comorbidity Index (n = 40), pain index (n = 65), ADL score (n = 13), and sPPF (n = 38). ^b^ Native Hawaiian/Pacific Islander/Mixed or Other.

**Table 2 ijerph-22-00704-t002:** Associated change in mean PA, SB, and sleep behavior outcomes for a 1-unit decrease in the intercept (baseline) and 0.3-unit per year decrease in slope of the short Performance-Based Physical Function score (sPPF) score trajectory ^a^ (N = 905).

Outcome	Individual Trajectory Features	Est ^b^	95% CI ^c^
**ActivPAL**			
**Sitting time (min/day) ^†^**	Intercept (baseline)	6.6	(1.2, 11.7)
	Slope	6.6	(−32.1, 52.0)
**Standing time (min/day) ^†^**	Intercept (baseline)	−2.5	(−7.0, 2.0)
	Slope	5.3	(−29.1, 38.7)
**Stepping time (min/day) ^†^**	Intercept (baseline)	−4.1	(−5.7, −2.6)
	Slope	−11.8	(−29.7, 0.1)
**Number of steps (count/day) ^†^**	Intercept (baseline)	−359	(−502, −226)
	Slope	−1180	(−2853, −185)
**Mean sitting bout duration (min/day) ^†^**	Intercept (baseline)	0.4	(0.05, 0.8)
	Slope	0.2	(−3.4, 3.3)
**ActiGraph**			
**Light activity (min/day) ^‡^**	Intercept (baseline)	−1.6	(−4.9, 1.6)
	Slope	−2.1	(−26.3, 21.9)
**Moderate-to-vigorous physical activity (min/day) ^‡^**	Intercept (baseline)	−4.4	(−6.1, −3.0)
	Slope	−15.7	(−35.6, −2.3)
**Self-Reported Sleep**			
**Time-in-bed (min/day)**	Intercept (baseline)	1.1	(−2.1, 4.3)
	Slope	6.0	(−15.0, 30.3)
**PROMIS Sleep Disturbance Score ^d^**	Intercept (baseline)	0.2	(−0.2, 0.5)
	Slope	1.6	(−0.6, 4.4)

^a^ A 0.3 unit decrease in slope per year translates to a decrease of 3 points on the sPPF score over 10 years. ^b^ Estimated from the regression coefficient from the multivariable linear regression model. The intercept (baseline) and slope exposures were derived from the linear mixed effects (lme) model fit to longitudinal data for sPPF collected during the 10 years prior to device wear and included in separate outcome models. The outcome and lme models were also adjusted for age in years, gender, body mass index kg/m^2^, education (16+ years vs. <16 years), current work for pay, CESD Score, CASI IRT, live alone vs. with others, self-rated health (4 levels, excellent/very good/good vs. fair/poor). ^c^ Confidence interval from percentile bootstrap algorithm with 1000 bootstrap replications. ^d^ PROMIS Sleep Disturbance Score has missing value n = 64. ^†^ Additionally adjusted for activPAL wear time. ^‡^ Additionally adjusted for ActiGraph wear time.

**Table 3 ijerph-22-00704-t003:** Associated change in mean PA, SB, and sleep behavior outcomes for a 1-unit increase in the intercept (baseline) and 0.4-unit-per-year increase in the slope of the Activity of Daily Living (ADL) score trajectory ^a^ (N = 905).

Outcome	Individual Trajectory Features	Est ^b^	95% CI ^c^
**ActivPAL Outcomes**			
**Sitting time (min/day) ^†^ **	Intercept (baseline)	8.8	(2.9, 13.9)
	Slope	35.0	(4.3, 65.0)
**Standing time (min/day) ^†^ **	Intercept (baseline)	−5.1	(−9.2, 0.01)
	Slope	−20.6	(−43.1, 4.4)
**Stepping time (min/day) ^†^ **	Intercept (baseline)	−3.7	(−5.2, −2.2)
	Slope	−14.4	(−24.2, −5.6)
**Number of steps (count/day) ^†^ **	Intercept (baseline)	−348	(−478, −225)
	Slope	−1372	(−2223, −638)
**Mean sitting bout duration (min/day) ^†^ **	Intercept (baseline)	0.8	(0.3, 1.3)
	Slope	3.5	(0.8, 6.2)
**ActiGraph**			
**Light activity (min/day) ^‡^ **	Intercept (baseline)	−0.9	(−4.1, 2.7)
	Slope	−8.0	(−23.9, 9.8)
**Moderate-to-vigorous physical activity (min/day) ^‡^ **	Intercept (baseline)	−3.3	(−5.0, −1.7)
	Slope	−13.0	(−22.6, −5.0)
**Self-Reported Sleep**			
**Time-in-bed (min/day)**	Intercept (baseline)	4.5	(1.0, 8.2)
	Slope	25.5	(6.5, 43.5)
**PROMIS Sleep Disturbance Score ^d^**	Intercept (baseline)	0.2	(−0.1, 0.6)
	Slope	1.0	(−0.9, 3.1)

^a^ A 0.4-unit increase in slope per year translates to an increase of 4 points on the ADL impairment summary score over 10 years. ^b^ Estimated from the regression coefficient from the multivariable linear regression model. The intercept (baseline) and slope exposures were derived from the linear mixed effects (lme) model fit to longitudinal data for ADL collected during the 10 years prior to device wear and included in separate outcome models. The outcome and lme models were also adjusted for age in years, gender, body mass index in kg/m^2^, education (16+ years vs. <16 years), current work for pay, CESD Score, CASI IRT, live alone vs. with others, self-rated health (4 levela, excellent/very good/good vs. fair/poor). ^c^ Confidence interval from percentile bootstrap algorithm with 1000 bootstrap replications. ^d^ PROMIS Sleep Disturbance Score is missing value for n = 64. ^†^ Additionally adjusted for activPAL wear time. ^‡^ Additionally adjusted for ActiGraph wear time.

## Data Availability

The data presented in this study cannot be made publicly available due to ethical and privacy restrictions. However, the datasets used in the current study are available upon reasonable request and execution of appropriate human subjects review and data sharing agreements by following the process described on the Adult Changes in Thought (ACT) website: https://actagingresearch.org (accessed on 21 April 2025).

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
