# Peer review of "Associations Between 10-Year Physical Performance and Activities of Daily Living Trajectories and Physical Behaviors in Older Adults"

_ijerph, 2025, doi:10.3390/ijerph22050704_

Round 1
Reviewer 1 Report
Comments and Suggestions for Authors
The manuscript addresses an important issue, but has several methodological and interpretative critical issues that limit its robustness. The sample analysed is not very diverse, with a predominance of white subjects, a high level of education and a generally better health status than the overall ACT cohort. This selection introduces a risk of bias in the generalisation of results, which should be discussed in more detail.
The assessment of physical function is based on the sPPF score, which excludes the balance test, a key component of the SPPB. This choice could reduce the sensitivity in detecting mobility impairment, affecting the observed associations with physical activity and sedentary behaviour. The exclusive use of self-reported sleep measures represents a further limitation, as it introduces possible biases in the assessment of sleep quality.
The associations found between decline in physical function and reduced motor activity are statistically significant, but of modest magnitude. The results suggest that relatively large changes in physical function are necessary to bring about clinically relevant changes in motor behaviour, an aspect that would merit more attention in the discussion. The absence of a clear association with sleep quality is interesting, but could be influenced by limitations in the measurement of sleep itself.
Overall, the study provides useful data, but needs a more critical analysis of the methodological limitations and greater clarity in the interpretation of the results.
Author Response
Comment 1: The manuscript addresses an important issue, but has several methodological and interpretative critical issues that limit its robustness. The sample analysed is not very diverse, with a predominance of white subjects, a high level of education and a generally better health status than the overall ACT cohort. This selection introduces a risk of bias in the generalisation of results, which should be discussed in more detail.
RESPONSE: We agree with the reviewer’s concerns here. Participants who agreed to wear activity monitors were, on average, younger and healthier than the broader ACT Study cohort. We present and discuss this in Supplemental Material eTable S1 and our limitations section (page 15, lines 500-507). In an effort to ensure that our findings are still broadly interpretable, particularly for older adults with frailty or other conditions that may be under-represented in the ACT sub-cohort who completed an accelerometer wear, we have implemented inverse probability weighting (IPW) in our outcome models as a sensitivity analysis. We now mention this in the last paragraph of the methods (page 5, lines 226-229). These findings are now discussed in the results section (page 9, lines 314-319; page 13, lines 397-400), and in the Supplemental Materials eTables S5 and S7 we present the new results tables with estimates and 95% confidence intervals, along with details of the IPW selection model. As described in the manuscript, results were very similar, with a qualitatively similar effect size and similar strength of evidence in terms of the magnitude of the p-value. We did see two borderline significant associations become non-significant for ADL trajectories and two borderline non-significant effects for sPPF trajectories become significant in the IPW results, which we do not interpret as important differences given the added variability in the IPW analysis.
Comment 2: The assessment of physical function is based on the sPPF score, which excludes the balance test, a key component of the SPPB. This choice could reduce the sensitivity in detecting mobility impairment, affecting the observed associations with physical activity and sedentary behaviour. The exclusive use of self-reported sleep measures represents a further limitation, as it introduces possible biases in the assessment of sleep quality.
RESPONSE: We agree with the reviewer on both points and have acknowledged the use of the sPPF, which excludes the balance task, and the self-reported sleep measures as limitations (page 15, lines 510-517; page 16, lines 518-522, respectively). However, these are the only measures available in this long-standing cohort and still offer valuable insights given the lack of prior literature examining the association of physical function and physical behaviors in this direction (i.e., as opposed to physical behaviors predicting function). Furthermore, in both cases, we believe any bias introduced would be towards the null. It is possible that additional and/or larger associations with physical behaviors exist that we were not able to detect in these analyses. Regarding the use of self-reported sleep measures, we agree that these are suboptimal. However, as the only data available representing sleep in this cohort, they offer a preliminary investigation into these relationships with historical function. The ACT cohort is currently collecting objective sleep data from wrist-worn actigraphy that we hope may allow future studies to explore how findings may differ with objective measures, but these data are not available for this current study. We have revised our discussion of these limitations throughout the discussion section – see edits on page 14-16 (lines 422-424, lines 454-471, lines 510-517, and lines 518-522,) to further underscore these limitations and point toward future directions for research that will address them.
Comment 3: The associations found between decline in physical function and reduced motor activity are statistically significant, but of modest magnitude. The results suggest that relatively large changes in physical function are necessary to bring about clinically relevant changes in motor behaviour, an aspect that would merit more attention in the discussion. The absence of a clear association with sleep quality is interesting, but could be influenced by limitations in the measurement of sleep itself.
RESPONSE: We agree with the reviewer’s suggestion to further expand upon the interpretation of findings between physical function and physical activity. Because the functional changes explored in our modeling results are relatively large, we have expanded this discussion to consider how we might interpret a smaller magnitude of functional change over the 10-year time period (page 14, lines 443-451). We believe this expanded discussion more fully underscores why we still believe these findings are of clinical significance and interest.
Regarding the reviewer’s comments about sleep, we agree that limitations in our measure could be at play and future research using objective measures is warranted. We note this in the text of our discussion (page 14-15, lines 463-472) and specifically discuss this in our limitations section (page 16, lines 518-522)
Comment 4: Overall, the study provides useful data, but needs a more critical analysis of the methodological limitations and greater clarity in the interpretation of the results.
RESPONSE: We appreciate the reviewer’s thoughtful suggestions to improve our presentation and interpretation of these findings. We have made revisions throughout our discussion and limitations sections, as noted above. We hope the revised discussion adequately acknowledges the limitations of this analysis and points to areas for future research.
The authors' full response letter with these and other reviewer comments and responses can be found in the attachment.

Reviewer 2 Report
Comments and Suggestions for Authors
The manuscript is very well written and understandable. The English language is fine.
The research topic on physical performance and activities in older adults is of high importance to combat diseases and promote health and well-being in our aging societies.
Important strengths include the longitudinal study design spanning 10 years, the detailed measurements, and the sample addressed.
The paper could nicely fit into the Special Issue Aging Strong: Promoting Exercise and Nutrition to Combat Frailty in Older Adults.
However, a number of issues are relevant to address:
The authors well acknowledge that there can be a bidirectional association between physical functioning and physical behaviors. But mostly, the behavior is treated as a predictor of PF. An inverse association (and circularity) seem most likely and need to be properly tested with e.g. cross-lagged models.
Some associations appear non-significanb. Has an a priori power analysis been conducted? Is the sample large enough for the kind of analysis performed? Are estimates reliable enough?
Bayesian modeling could help to ease interpretation of the null effects, i.e. to be more sure that the conclusion is valid that there are no effects. Maybe the effects are just small. Maybe consider reformulating the main conclusions in this regard.
The sample may allow building subgroups to investigate differential association patterns between e.g. women vs. men, young-old vs. old-old, low vs. high initial health status / socioeconomic position / frailty etc.
It could be better highlighted the novelty, e.g. how in detail are existing conceptual models on aging and health are advanced.
The practical implications could be better highlighted with more detailed examples from everyday life. For instance, regarding which of the predictors need to be addressed first?
Author Response
The manuscript is very well written and understandable. The English language is fine. The research topic on physical performance and activities in older adults is of high importance to combat diseases and promote health and well-being in our aging societies. Important strengths include the longitudinal study design spanning 10 years, the detailed measurements, and the sample addressed. The paper could nicely fit into the Special Issue Aging Strong: Promoting Exercise and Nutrition to Combat Frailty in Older Adults. However, a number of issues are relevant to address:
RESPONSE: We appreciate the reviewer’s comments about the manuscript’s strengths and thoughtful suggestions to improve it. We individually address each of the issues raised below.
- The authors well acknowledge that there can be a bidirectional association between physical functioning and physical behaviors. But mostly, the behavior is treated as a predictor of PF. An inverse association (and circularity) seem most likely and need to be properly tested with e.g. cross-lagged models.
RESPONSE: We agree with the reviewer’s comment – the relationship between physical function and physical behaviors is complex. There are likely many methodological approaches that could be undertaken to explore that complexity. The cross-lagged modeling framework the reviewer suggests is interesting, but we don’t believe our data are appropriate for that modeling approach at present, particularly given the debate in the literature about that appropriate application for such models (Lucas 2023). We are still accumulating additional prospective data on function and physical behaviors and hope to more fully explore the bidirectionality of this relationship in future work when more follow-up data are available. However, in the meantime, we believe exploring the relationship in a single direction is appropriate, given the dearth of prior literature exploring this direction.
Lucas, R. E. (2023). Why the cross-lagged panel model is almost never the right choice. Advances in Methods and Practices in Psychological Science, 6(1), Article 25152459231158378. https://doi.org/10.1177/25152459231158378
- Some associations appear non-significant. Has an a priori power analysis been conducted? Is the sample large enough for the kind of analysis performed? Are estimates reliable enough?
RESPONSE: This analysis was exploratory, and as such, we feel the width of the 95% confidence intervals is the more relevant measure of the reliability, providing the range of effects consistent with our data, rather than over-emphasizing the formal testing framework. In terms of non-significant results, we edited the Discussion section to better reflect this interpretation, and limitation, that with our sample size we could not distinguish between null effects or smaller effects needing larger sample sizes to be reliably detectable. See pages 14-15, lines 454-457, 468-472, lines 507-510, and lines.
- Bayesian modeling could help to ease interpretation of the null effects, i.e. to be more sure that the conclusion is valid that there are no effects. Maybe the effects are just small. Maybe consider reformulating the main conclusions in this regard.
RESPONSE: We agree that the null effects observed here are not definitive and it is possible effects exist that were not detectable here. We have revised our discussion to more clearly acknowledge this possibility. See pages 14-15, lines 454-457, 468-472.
- The sample may allow building subgroups to investigate differential association patterns between e.g. women vs. men, young-old vs. old-old, low vs. high initial health status / socioeconomic position / frailty etc.
RESPONSE: While we agree that future exploration of differential relationships in subgroups of interest is worthwhile, we do not believe that we are adequately powered to pursue these analyses in our sample. Furthermore, such subgroup analyses would add many additional comparisons that would be best done with corrections for multiple comparisons, which would further lower power. We have added a sentence in the limitations section to acknowledge this and encourage future research to examine these potential differential associations. See page 15, lines 507-510.
- It could be better highlighted the novelty, e.g. how in detail are existing conceptual models on aging and health are advanced.
RESPONSE: Thank you for highlighting this opportunity to more clearly articulate the contribution of these findings to our broader knowledge of older adult health. We have added a new paragraph to our discussion that addresses how these findings further our understanding of older adult functional health and point to practical implications of these findings. This paragraph reads:
“Preservation of physical function is critical to the preservation of independence and quality of life as we age. However, to design and optimally time the delivery of health promotion and preservation strategies for older adults to protect physical function, it is imperative to fully characterize its bidirectional link with physical behaviors. In other words, we must understand not just how physical behaviors drive physical function, but also how physical function declines drive activity. In the context of prior literature, the findings from this analysis support the bidirectional nature of the relationship of physical function and physical behaviors, underscoring that the two likely work synergistically to either promote or deteriorate health and independence in older age. In a practical sense, these findings highlight the need for intervention to improve and maintain physical activity and reduced sedentary behaviors early in the life course as well as at times of transition in functional capacity (e.g., after an injury or change in health status). As prior literature has underscored, higher PA and lower SB are protective of future physical function. While age- and morbidity-related declines in physical function will not all be preventable, early intervention in physical activity may maximize functional resilience over time and prevent spiraling declines in future activity and behaviors.”
- The practical implications could be better highlighted with more detailed examples from everyday life. For instance, regarding which of the predictors need to be addressed first?
RESPONSE: We appreciate this suggestion as well. Please see our response to the prior comment, which addresses this suggestion as well, outlining practical implications of these findings for future interventions to promote older adult functional health and physical behaviors.
The authors' full response letter with these and other reviewer comments and responses can be found in the attachment.

Reviewer 3 Report
Comments and Suggestions for Authors
The authors present an interesting article titled: “Associations Between 10-year Physical Performance and Activities of Daily Living Trajectories with Behaviors in Older Adults”. The study addresses a highly relevant topic, which is becoming increasingly significant due to demographic changes.
Comments and suggestions:
- While reading the introduction, the question arises as to what the “Adult Changes in Thought (ACT) cohort” refers to. A reference to the detailed explanations in the Materials and Methods section could be helpful for clarity.
- The introduction concludes with a thorough presentation of the study's objectives and hypotheses.
- The Materials and Methods section is detailed and provides extensive information on the study population, instruments used, and statistical methods.
- The results are presented clearly and are well supported by numerous figures and tables. Additional information is available in the supplementary file.
- When analyzing the patient characteristics, it becomes evident that, despite a mean age of 77.6 years, the study population appears to be relatively healthy. Notably, 93% of participants perceive their health as at least good (with 19.7% even rating it as excellent), and the Charlson Comorbidity Index remains low. This suggests a potential underrepresentation of age-related conditions such as frailty. Given the clinical and prognostic relevance of comorbidities and conditions like frailty, these aspects warrant further consideration and discussion.
Overall, the manuscript is well-written and addresses an interesting and relevant topic. Minor adaptations could further improve the manuscript.
Author Response
The authors present an interesting article titled: “Associations Between 10-year Physical Performance and Activities of Daily Living Trajectories with Behaviors in Older Adults”. The study addresses a highly relevant topic, which is becoming increasingly significant due to demographic changes.
RESPONSE: Thank you for your comments and thoughtful suggestions to improve this manuscript. Responses to the concerns raised can be found below.
Comments and suggestions:
- While reading the introduction, the question arises as to what the “Adult Changes in Thought (ACT) cohort” refers to. A reference to the detailed explanations in the Materials and Methods section could be helpful for clarity.
RESPONSE: Thank you for pointing out this opportunity to improve clarity. We have revised our first mention of the ACT cohort in the introduction to provide a brief orientation and point readers to the methods section for more details.
- The introduction concludes with a thorough presentation of the study's objectives and hypotheses.
RESPONSE: Thank you for your comment.
- The Materials and Methods section is detailed and provides extensive information on the study population, instruments used, and statistical methods.
RESPONSE: Thank you for your comment.
- The results are presented clearly and are well supported by numerous figures and tables. Additional information is available in the supplementary file.
RESPONSE: Thank you for your comment.
- When analyzing the patient characteristics, it becomes evident that, despite a mean age of 77.6 years, the study population appears to be relatively healthy. Notably, 93% of participants perceive their health as at least good (with 19.7% even rating it as excellent), and the Charlson Comorbidity Index remains low. This suggests a potential underrepresentation of age-related conditions such as frailty. Given the clinical and prognostic relevance of comorbidities and conditions like frailty, these aspects warrant further consideration and discussion.
RESPONSE: We agree with the reviewer’s concerns here. Participants who agreed to wear activity monitors were, on average, younger and healthier than the broader ACT Study cohort. We present and discuss this in Supplemental Material eTable S1 and our limitations section (page 15, lines 500-507). In an effort to ensure that our findings are still broadly interpretable, particularly for older adults with frailty or other conditions that may be under-represented in the ACT sub-cohort who completed an accelerometer wear, we have implemented inverse probability weighting in our outcome models as a sensitivity analysis. We now mention this in the last paragraph of the methods (page 5, lines 226-229). These findings are now discussed in the results section (page 9, lines 314-319; page 13, lines 397-400), and in the Supplemental Materials eTables S5 and S7 we present the new results tables with estimates and 95% confidence intervals, along with details of the IPW selection model. As described in the manuscript, results were very similar, with a qualitatively similar effect size and similar strength of evidence in terms of the magnitude of the p-value. We did see two borderline significant associations become non-significant for ADL trajectories and two borderline non-significant effects for sPPF trajectories become significant in the IPW results, which we do not interpret as important differences given the added variability in the IPW analysis.
Overall, the manuscript is well-written and addresses an interesting and relevant topic. Minor adaptations could further improve the manuscript.
RESPONSE: Thank you for your comment and the previously addressed suggestions for improvement.
The authors' full response letter with these and other reviewer comments and responses can be found in the attachment.

Reviewer 4 Report
Comments and Suggestions for Authors
The present study aimed to study bidirectional relationship between physical function and physical behaviors, specifically physical activity, sedentary behavior, and sleep in older adults aged 65 and above. The researchers examined how 10-year trajectories of physical performance (sPPF) and self-reported impairment in activities of daily living (ADL) predicted patterns of these physical behaviors in later life.
Despite the importance of this theme, there are some concerns related to study conceptualization. The problem of the study should be deeply explored. A descriptive analysis about the prevalence of the PA/sedentary behavior and sleep quality as problematic in this population will enhance the study's pertinence. Moreover, the gap that this study will fill isn't enough. The authors only identified that “The existing literature on the relationship between sleep and physical function is predominantly cross-sectional, and findings are mixed.” We need a really strong justification for this study.
Author Response
The present study aimed to study bidirectional relationship between physical function and physical behaviors, specifically physical activity, sedentary behavior, and sleep in older adults aged 65 and above. The researchers examined how 10-year trajectories of physical performance (sPPF) and self-reported impairment in activities of daily living (ADL) predicted patterns of these physical behaviors in later life.
RESPONSE: Thank you for your careful consideration of our manuscript. Responses to the concerns raised can be found below.
- Despite the importance of this theme, there are some concerns related to study conceptualization. The problem of the study should be deeply explored. A descriptive analysis about the prevalence of the PA/sedentary behavior and sleep quality as problematic in this population will enhance the study's pertinence.
RESPONSE: Thank you for highlighting this opportunity to improve our justification for this work. As the reviewer suggests, we have added some additional background information about the prevalence of physical activity, sedentary behavior, and sleep disturbance in the older adult population. Please see page 2, lines 43-51.
- Moreover, the gap that this study will fill isn't enough. The authors only identified that “The existing literature on the relationship between sleep and physical function is predominantly cross-sectional, and findings are mixed.” We need a really strong justification for this study.
RESPONSE: Thank you for pointing out the need for added clarity and specificity about the gap this study’s findings fill. We have added additional language to our introduction (page 2, lines 43-51, 82-87) and a new paragraph to our discussion (page 15, lines 473-488) that underscores the importance of fully characterizing the bidirectional nature of the relationship between physical function and physical behaviors and points to some practical implications of these findings for older adult health promotion efforts. We also edited our conclusion to emphasize the importance and applicability of these findings (page 16, lines 535-540).
The authors' full response letter with these and other reviewer comments and responses can be found in the attachment.

Round 2
Reviewer 1 Report
Comments and Suggestions for Authors
I thank the authors for their detailed and thoughtful responses. The revised manuscript adequately addresses the main concerns raised in the first round. The use of inverse probability weighting is appropriate and well described, and the discussion of sample selection and potential bias has been expanded. The limitations related to the use of the sPPF without the balance task and the reliance on self-reported sleep data are clearly acknowledged and justified based on the available data. The discussion of the small magnitude of the associations and their clinical interpretation is now more balanced and informative. I invite the authors to consider briefly mentioning in the main text the role of IPW in mitigating selection bias, currently discussed mainly in the supplementary material. It might also be useful to explicitly state that the absence of the balance task in the sPPF might have led to an underestimation of associations, especially with low-intensity tasks. Overall, these are minor points and the manuscript is now much improved and suitable for publication after a minor revision.
Author Response
COMMENT 1: I thank the authors for their detailed and thoughtful responses. The revised manuscript adequately addresses the main concerns raised in the first round. The use of inverse probability weighting is appropriate and well described, and the discussion of sample selection and potential bias has been expanded. The limitations related to the use of the sPPF without the balance task and the reliance on self-reported sleep data are clearly acknowledged and justified based on the available data. The discussion of the small magnitude of the associations and their clinical interpretation is now more balanced and informative.
RESPONSE: Thank you for acknowledging our previous revisions. We appreciated your constructive feedback and suggestions.
COMMENT 2: I invite the authors to consider briefly mentioning in the main text the role of IPW in mitigating selection bias, currently discussed mainly in the supplementary material. It might also be useful to explicitly state that the absence of the balance task in the sPPF might have led to an underestimation of associations, especially with low-intensity tasks. Overall, these are minor points and the manuscript is now much improved and suitable for publication after a minor revision.
RESPONSE: Thank you for these additional suggestions to improve our manuscript. We have expanded our discussion of the IPW and its role in addressing selection bias. We also now include an AUC statistic that directly informs the success of our selection models in addressing the reasons for differential missingness. Please see added text in the Methods, Results, and Discussion sections (pg 5, lines 225-228; pg. 9, lines 315-319; pg. 13, lines 400-402; pg. 15, lines 508-514). Additionally, we now explicitly note that the absence of the balance task (particularly when replaced with the grip strength task) is likely to have led to underestimation in associations, particularly for lower intensity movement outcomes. Please find this revision in the Discussion (pg. 14, lines 427-430).
Reviewer 4 Report
Comments and Suggestions for Authors
Dear authors,
thank you for addressing my comments.
Author Response
Thank you for your careful review.